# OpenReview forum: "I’m Absolutely Right: Self-Attribution Bias in LLM Agents"
_ICML.cc/2026/Conference — Submitted to ICML 2026_

### Official Review · Reviewer_1ys8 · 2026-03-04

**Soundness:** 3
**Presentation:** 1
**Significance:** 2
**Originality:** 2
**Overall Recommendation:** 3
**Confidence:** 3

**Summary:**

The paper identifies self-attribution bias in agentic systems, showing that language-model monitors are more likely to rate actions as correct or safe when those actions appear in their own prior assistant turns rather than in a separate user turn. Experiments across coding and tool-use tasks reveal that this bias leads to underreporting of risky or incorrect actions, suggesting that standard evaluation setups may overestimate monitor reliability in deployment.

**Compliance With Llm Reviewing Policy:**

Affirmed.

**Key Questions For Authors:**

1. What is the difference between Same Turn/Previous Turn/Baseline and On-policy/Off-policy?

2. This study finds that implicit structures are more prone to bias than explicit ones. This suggests that models tend to favour instruction-following, being more sensitive to explicit cues and immediately recognising them. Conversely, implicit structures, by drawing upon context, create a plausible illusion.

3. Can evaluations based on other metrics maintain consistent conclusions when core metrics are measured on a scale of 0–10?

**Limitations:**

yes

**Strengths And Weaknesses:**

Strengths:
1. This work is highly intriguing, analysing whether the model exhibits the same overconfidence during the current dialogue and subsequent sobering-up behaviour as humans do.

2. This work demonstrates that agents may exhibit this phenomenon in on-policy and same-turn contexts due to some form of consistency or attribution. This may indicate that context exerts a fundamental influence on an agent's decision-making.

Weaknesses:
1. Could context become a key factor in shaping an agent's evaluation bias? If I understand correctly, this paper sets up identical/prior-round experiments depending on whether the evaluation request originates from the user or the assistant. Furthermore, on-policy and off-policy designs are employed to analyse whether actions stem from the agent itself. Thus, when initiating evaluation again within an exceptionally lengthy contextual dialogue, can the agent maintain this bias?

2. It is difficult to assess whether the differences between on-policy and off-policy learning stem from self-attribution or distribution variance.

3. This paper focuses on the self-attribution of intelligent agents, subsequently demonstrating that explicit attribution fails to hold. This essentially proves that self-attribution is inherently weak. This phenomenon appears more akin to contextual consistency or attention anchoring.

4. Considering the autoregressive paradigm, if I understand correctly, same-turn requires the model to simultaneously predict actions and evaluation. Clearly, agents will confirm their actions along semantic directions, leading to the phenomenon proposed in this paper. Furthermore, llm-as-judge appear to be common employed by adopting larger, more generalised models, rather than self-evaluation.

5. The paper emphasises that off-policy evaluation overestimates risk, yet fails to systematically test viable alternative designs.

6. The appendix appears to be single-column, and Table 1 has display issues. Page 14, lines 764–766 contain citation issues.

---

> ### Author Rebuttal · Authors · 2026-03-31
>
> W1:
> To address concerns on long context, long horizon, very multi turn realistic agentic tasks, we run this on agent-swe-bench, where we see a similar on-policy self-attribution bias, see the figure https://figshare.com/s/9283585a0a1bb9ef32dc?file=63332530 .  Self-attribution bias in an agentic SWE-bench setting is arguably more alarming because that is much closer to a real deployed use case.
>
> In addition, we have a breakdown of absolute correctness ratings (0–10) for the original, single step swe bench: https://figshare.com/s/9ab702ceacb108145635
>
>
> W2/W3:
> If on-policy/off-policy differences reflected distributional variance in artifact quality rather than self-attribution, we would expect inflated ratings whenever a model evaluates on-policy artifacts, regardless of prompt format. However, our explicit attribution ablation shows this is not the case: on-policy artifacts evaluated under baseline prompt formats (new context, artifact in user turn) receive ratings similar to off-policy baselines. The inflation appears specifically and only under self-attributed prompt formats applied to on-policy artifacts.
>
> Furthermore, with n=300 per condition and the small error bars visible throughout our figures, distributional noise is unlikely to account for the consistent effect sizes we observe. Finally, our cross-model heatmaps (Figure 6) show the effect concentrated on the diagonal, models inflating ratings specifically for their own outputs, which would not be predicted by a distributional variance account, since related models generating similarly-distributed artifacts do not show comparable inflation off-diagonal.
>
>
> W4:
> We largely agree that same-turn evaluation can be biased because the model is predicting action and rating in the same turn and thus could play the same consistent character. However we also show self-attribution bias across turns (in the previous turn setting). We also don’t think these conclusions were overdetermined, since in other contexts language models sometimes criticize themselves: for example reasoning models often do backtracking, and LLMs often admit to lies when directly confronted about it.
>
> Self-evaluation is not unrealistic, developers use claude code to write code and organizations use claude code to review PRs. Additionally, previous work (including from frontier AI companies) has studied this kind of monitoring strategies (see https://openai.com/index/how-confessions-can-keep-language-models-honest/).
>
>
> W5:
> We provide alternative designs:
> * The baseline prompt formats, which almost entirely mitigates this self-preference (see Figure 7, and see new figure with new cross model heatmap  https://figshare.com/s/9ab702ceacb108145635 , which we will add to the paper)
> * Using a different model, which partially mitigates the effect (see Figure 6)
> * Using more reasoning tokens, which partially mitigates this effect (Figure 5)
> We will summarize these mitigations in a new subsection of the discussion.
>
> W6:
> Thank you for noticing these issues. We will fix them.
>
> Q1:
> Same-turn/previous-turn/baseline is the prompt format describing how the artifact is presented to the model at evaluation time, regardless of who generated it. On-policy/off-policy describes the origin of the artifact, whether the evaluating model itself generated it during the current interaction, or whether it is a fixed artifact generated by another model or human and prefilled in. These differences are explained in Section 3.1.
>
> Q2:
> We indeed find that implicit attribution matters more than explicit attribution, as discussed in Section 4.3.
>
> Q3:
> We use different ratings scales, for example our PR approval results were simply asking whether the model believes the PR is good to merge. Computer use was measured on a scale of 0-100 and normalized. Other settings use the 0-10 rating scale. Therefore we do not believe that our conclusions are sensitive to the rating scale used.
>
> If you are convinced, we would appreciate it if you might consider revising your review.

---

> > ### Author Rebuttal · Reviewer_1ys8 · 2026-04-02
> >
> > Thank you for your reply!
> >
> > * Does the statement in W5 that ‘using different models can partially mitigate this effect’ prove that self-assessment is unreliable? Therefore, the response to W4 is contradictory.
> >
> > * If ‘the baseline prompt format has almost entirely mitigated this self-bias’, then it is difficult to quantify the stability and reliability of the self-bias assessment.
> >
> > Therefore, I will maintain my initial score.

---

> > > ### Author Response · Authors · 2026-04-02
> > >
> > > Thank you for these follow-up questions. There is no contradiction between our W4 and W5 responses because they address different claims. W4 says self-monitoring is realistic and used in practice, W5 says using a different monitor format mitigates the bias.
> > >
> > > Our W4 response makes the following claim: self-monitoring is realistically deployed in practice and therefore worth studying (e.g. Claude Code reviewing its own PRs), defending the relevance of our experimental setup. It was defending the relevance and realism of your experimental setup, we are not defending self-monitoring as a reliable or good practice.
> > >
> > > Our W5 response makes a separate claim: when self-monitoring is used, it introduces a measurable failure mode and using a different monitor format partially mitigates the bias. This is a practical recommendation that self-monitoring has a failure mode.
> > >
> > > The baseline prompt format does largely mitigate the bias, but this requires architectural changes to deployed pipelines, specifically breaking KV cache sharing with the main agent and restructuring the monitoring call, which is why developers sometimes use the cheaper follow-up format in the first place (moving the artifact to a fresh context requires a separate API call, losing the computational benefits of continuing in the same context). Our central finding is that off-policy evaluations do not reveal this bias, and a developer benchmarking their monitor on fixed artifacts would observe AUROC of 0.99 and conclude their monitor is reliable, while the on-policy AUROC is 0.89. The bias is difficult to detect because the evaluation methodology masks it.
> > >
> > > off-policy evaluation (the benchmark) overestimates monitor quality relative to on-policy performance (the real world). The developer sees 0.99 in their evaluation, but gets 0.89 in practice. The paper demonstrates a failure mode that is easy to fix once you know about it but hard to detect in the first place, a meaningful contribution to document.
> > >
> > > We hope this clarifies the distinction and would appreciate it if you would consider positively updating your scores.

---

### Official Review · Reviewer_8Eht · 2026-03-10

**Soundness:** 3
**Presentation:** 3
**Significance:** 4
**Originality:** 3
**Overall Recommendation:** 4
**Confidence:** 3

**Summary:**

The article explores a failure mode in monitoring systems based on LLM known as self-attribution bias, which is a judgment of an action as more favorable by language models in the case where they assume the source of the action to be themselves as opposed to being the result of a different source. The paper provides an analysis of how conversational structure (e.g., testing an action produced in the same or previous assistant turn) implicitly assigns authorship and causes biased assessments.

In general, the key purpose of this manuscript is related to finding and determining biases in LLM self-monitoring systems in agentic systems like code assistants or tool-using agents. In general, the inherent contribution of this manuscript refers to the fact that the same outputs are most likely to be considered as safer or more correct when implicitly assigned to the model of evaluation.

The authors compare this phenomenon in various spheres such as code correctness (SWE-bench), computer-use risk, code security risk, and question-answering tasks with 10 frontier language models. The findings indicate that the self-attribution bias is most pronounced in on-policy situations in which models review actions they have just received, and that the off-policy evaluations of statistics can greatly exaggerate the reliability of monitoring.

**Compliance With Llm Reviewing Policy:**

Affirmed.

**Key Questions For Authors:**

The experimental research deals with minimal interaction environments. What is the behavior of self-attribution bias in multi-specific agentic processes that go through extended multi-step processes?

The article demonstrates that the bias is multiplied by the on-policy evaluations. Have the authors measured the frequency of such conditions in actual agent deployments in the real world?

Are some prompting tactics or architectural designs lessening self-attribution bias or much of it has to do with existing paradigms of LLM training?

Have the authors taken into consideration mechanistic analyses (e.g., probing representations or attribution patterns) to gain a better insight into the cause of this bias?

Does ensemble or cross-model evaluation mitigate the bias by making models not evaluate their own results?

**Limitations:**

The authors point out various limitations of the study such as the relatively simplified prompts and datasets and the evaluation in long multi-step agentic interactions were not done. They recognize that the scale of self-attribution bias might be different in more realistic long-horizon deployment contexts and that their experiments are not comprehensive in that regard. Also in the paper, it is observed that in certain instances not all evaluation conditions can be considered on-policy, and therefore the effect size observed can be influenced by this. In general, the article offers a credible discussion of these limitations and their implications of the study of the results.

**Strengths And Weaknesses:**

Strengths

Soundness

The article has a very clear phenomenon it is set out to describe (self-attribution bias) and therefore suggests a well-designed experimental design to quantify it.

The research involves the use of various data sets such as SWE-bench, computer-use risk scenarios as well as MMLU-style tasks, which covers a wide range of empirical.

In the analysis, the difference between on-policy and off-policy monitoring settings is identified, and it is necessary to analyze real-world deployment behavior.

Presentation

The paper is structured well, and it has clear methodology, datasets, results, and implications sections.

The impact of the bias can be explained in terms of figures that describe rating changes and risk inaccuracy.

The theory of implicit vs explicit attribution is very elucidated.

Significance

The article is relevant to the field of LLM safety and evaluation, specifically to the agentic systems that are based on self-monitoring.

The results point to the possibility of a discrepancy between benchmark assessment and the actual performance of the system in the workplace.

Originality

The presence of the self-attribution bias of the concept of the LLM monitoring is a new framing of the failure in evaluation.

The evaluation relates the behavioral biases that may be exhibited by humans with the biases that may form in the process of language model evaluation.

Weaknesses

Soundness

The research is primarily based on timely experiment and observational research; a more detailed mechanistic research of the reasons why models display the bias would help make the work stronger.

Certain review cases are still reasonably simplistic in relation to real-life agentic workflows of extended prospects.

Presentation

There are quite a few parts that have a lot of detailed methodology that could be more concisely summarized on a higher level.

Certain details of the experimental set-up (e.g. prompt structures, scoring calibration) might be described in more detail.

Significance

Although the paper illustrates the existence of the bias, it offers scanty practical mitigation measures to practitioners.

Originality

The study is a continuation in the current literature on the bias and self-preference in the evaluation of LLM, although the framing is novel, certain conceptual elements overlap with the previous literature on the biases of LLM-as-a-judge.

---

> ### Author Rebuttal · Authors · 2026-03-31
>
> W1:
> To address concerns on realistic agentic tasks, we run this on agent-swe-bench, see figure https://figshare.com/s/9283585a0a1bb9ef32dc  which shows attention shift away from artifact content and toward the turn marker under self-attribution, evidence the model is responding to the ownership signal itself rather than anything about the code. We will also add analysis of attention patterns here https://figshare.com/s/418b244ed5472f6f746a  (as suggested by reviewer 8Eht).
>
> W2:
> We will revise Section 3 to open with a concise 2-3 sentence summary of the experimental logic before the technical details e.g., explicitly stating upfront that we vary exactly two things (attribution format and action origin) while holding the artifact fixed  so readers can orient themselves before encountering the prefilling and turn-structure mechanics.
>
> Since submission we reorganized the appendix and added full prompts, rubrics, and scoring details to the appendix, which we hope satisfies the request for experimental specificity.
>
> W3:
> In the paper we discuss the following mitigations:
> 1. Use a fresh context for monitoring.
>       * Don't use follow-up question prompt formats for monitoring.
>       * Simply moving the artifact from an assistant turn to a user turn in a new conversation largely eliminates the bias. The on-policy/off-policy AUROC gap (0.89 vs 0.99) supports this.
> 2. Use a different model as a monitor. Use off-policy evaluation (pull the transcript and artifact into a fresh context). See cross model heatmaps in Figure 6.
> 3. Be skeptical of static off-policy evaluations and instead use on-policy evaluations.
> Increasing the reasoning also reduces the effect slightly (see figure 5)
>
> We will summarize these mitigations in a new subsection of the discussion.
>
> W4:
> The two most similar works are Wataoka et al. (2410.21819) and Chen et al. (2502.06655). Wataoka et al. study self-preference in the context of benchmark contamination specifically, whether a model favors its own style when choosing between two alternative outputs in a pairwise comparison. Chen et al. study model bias in Agents-as-an-Evaluator pipelines, again motivated by benchmark contamination, where their "model bias" is a between-model distributional effect arising from shared stylistic tendencies across model families. Both works operate entirely in static, off-policy settings and are motivated by benchmark fairness concerns.
>
> Our paper differs from both along the following axes:
> (1) The phenomenon is different. We study how the rating of a single fixed artifact changes solely due to implicit authorship attribution.
> (2) The bias is largely absent off-policy and emerges specifically on-policy. This would be invisible to both prior works, which never generate actions on-policy. This means standard static evaluations actively mask the failure.
> (3) Prior work's implications concern leaderboard reliability. Our concern is whether a deployed agent will approve its own unsafe actions with documented cases of 9→0/10 risk reversals on document disclosure tasks and 5× higher approval rates for prompt-injected code.
> (4) We are the first to connect this failure mode to AI control and alignment concerns, showing that collusion-like monitor failures can arise in non-scheming systems
>
> Building on broader literature on LLM evaluation bias while introducing a novel phenomenon and implications for AI alignment is incremental research.
>
> Q1/Q2:
> To address concerns on long context, long horizon, very multi turn realistic agentic tasks, we run this on agent-swe-bench, see the figure https://figshare.com/s/9283585a0a1bb9ef32dc?file=63332530 .  We see self-attribution bias in an agentic SWE-bench setting is arguably more alarming because that is much closer to a real deployed use case.  In addition, we have a breakdown of absolute correctness ratings (0–10) for the original, single-step swe bench: https://figshare.com/s/9ab702ceacb108145635
>
> Q3:
> We tried different prompt variations.(1) trying to vary authorship, (2) trying to vary prompt structure.
> For authorship (1)
> Human
> No authorship
> Same model
> Another model
>
> For prompt structure (2):
> We also changed the baseline prompt so that the monitoring occurs in a different structural positions.
>
> Across these variations, we find that not using a previous or same turn prompt format is the most important way to avoid self-attribution bias.
>
> Q4:
> We analyzed attention patterns in the code harmfulness settings on Gemma 27B. The baseline prompt resulted in more attention being paid to the artifact content, see https://figshare.com/s/418b244ed5472f6f746a?file=63332224. We will add this figure and analysis in the Appendix.
>
> Q5:
> It doesn’t mitigate the bias but it does significantly reduce it, see off diagonals in figure 6.
>
> We are glad to incorporate all of the changes above and hope they address your concerns. If you feel the revision is satisfactory, we would be grateful if you would consider updating your score accordingly.

---

> > ### Author Rebuttal · Reviewer_8Eht · 2026-04-02
> >
> > Thanks to the comprehensive and positive refutation. The further experiments on more realistic agentic environments (e.g., Agent-SWE benchmarks) are effective in addressing issues related to simplistic evaluation situations and prove that the bias of self-attribution is present in the more realistic multi-step contexts. The presence of attention-based analyses offers helpful preliminary information on the mechanism behind the model, especially the sensitivity of the model to implicit cues of authorship, as opposed to the content of the artifact.
> >
> > The increased coverage of mitigation measures, including the isolation of context, cross-model monitoring, and restructuring evaluation prompts, contribute to the work being practically relevant to a great extent. Issues of reproducibility and presentation are further clarified by the clarification of experimental setup, including further prompt and scoring information.
> >
> > Moreover, the location of the work in respect of previous literature is now more evident, particularly the on-policy character of the bias witnessed versus previous off-policy assessment researches.
> >
> > In general, the rebuttal counters the key issues raised in the review well and supports both empirical and practical contributions of the article.

---

> > > ### Author Response · Authors · 2026-04-02
> > >
> > > Thank you, we're glad we were able to address your concerns. If you feel everything has been fully resolved, we'd truly appreciate it if you'd consider improving your score.

---

### Official Review · Reviewer_3bHc · 2026-03-12

**Soundness:** 3
**Presentation:** 3
**Significance:** 3
**Originality:** 3
**Overall Recommendation:** 5
**Confidence:** 3

**Summary:**

This paper demonstrates a concerning form of bias in LLM agents. They find that self-monitoring agents are likely to be biased in favour of their own outputs in a range of monitoring situations (e.g. code generation, agent computer use tasks, MCQs). This effect is not observed when monitoring identical outputs appearing to be from other models. This is concerning because monitoring agents are usually evaluated in settings where self-attribution bias would not occurr, but then deployed in settings where it would.

**Compliance With Llm Reviewing Policy:**

Affirmed.

**Key Questions For Authors:**

None

**Limitations:**

Yes

**Strengths And Weaknesses:**

Strengths:
- The submission is technically sound; the claims are well supported by experimental results. Multiple possible confounding factors are controlled for. The claims are tested on a realistic and varied set of tasks.
- The presentation is, for the most part, good; the theoretical motivation, methodology, and results were easy to follow
- The paper addresses an important problem that may occurr in production, and may be missed in testing due to the reasons explained in the paper.
- While it is similar to some prior work regarding colluding and self-preferencing AI, the phenomenon it examines is sufficiently original.

Weaknesses:
- The paper stops short of trying to explain why this phenomenon occurrs, and mentions that future work could seek to "understand the mechanistic causes of self-attribution bias". However, in the conclusion it seems to hint at a possible explanation, namely that LLMs trained for coherence might cause self-attribution bias: 'as LLMs are trained to behave like coherent agents, their coherence can distort evaluations'. This sounds plausible, but it would be better for the authors to either put this forward as a possible explanation in the discussion, and unpack it a bit more substantially, or else avoid speculating on it in the conclusion.
- I find the terms 'on-policy' and 'off-policy' to be unintuitive. I understood them when I read the definitions in the paper, but it would be better to have more intuitive terms. Perhaps 'within-interaction' vs 'across-interaction', 'within-session', or similar, would be more intuitive.

---

> ### Author Rebuttal · Authors · 2026-03-31
>
> W1:
> We will address this by moving the hypothesis into the Discussion and expanding it substantially. Concretely, we will argue that RLHF and instruction tuning may reward behavioral coherence, and select for agents that act consistently and don't contradict their prior outputs.
>
> W2:
> We are using words from the RL literature, where on-policy refers to samples which are generated by the policy itself, while off-policy refers to samples generated by another policy, which matches our usage. However, we agree this could be unintuitive. We can change the terms on policy and off policy to “generated within interaction” and “generated across interaction” The “generated” is important to show that the model generated the artifact.
>
> We hope these clarifications resolve the ambiguities and give you greater confidence in evaluating the work.

---

> > ### Author Rebuttal · Reviewer_3bHc · 2026-04-01
> >
> > These changes would fully resolve the weaknesses mentioned in the review.

---

### Official Review · Reviewer_n3xK · 2026-03-13

**Soundness:** 2
**Presentation:** 3
**Significance:** 2
**Originality:** 3
**Overall Recommendation:** 3
**Confidence:** 4

**Summary:**

This paper identifies and defines Self-Attribution Bias, a phenomenon where Large Language Models (LLMs) evaluate their own actions as significantly more correct or less risky compared to identical actions presented without attribution or attributed to others. In high-stakes scenarios like coding agents (SWE-bench) and computer use (tool use), this bias leads monitors to approve unsafe code (e.g., prompt injections) or risky actions.

**Compliance With Llm Reviewing Policy:**

Affirmed.

**Key Questions For Authors:**

- What is the main new novelty compared with previous Self-Attribution Bias/Preference Bias/Model Bias works？

**Strengths And Weaknesses:**

S:
- The paper highlights a fundamental flaw in how AI agents are currently evaluated. It demonstrates that static, off-policy benchmarks likely overestimate the reliability of self-monitoring systems in real-world agentic deployments.

W:
- While the paper addresses agents, the experiments mostly involve single-turn or short-horizon generate-and-evaluate loops. The complexity of long-horizon tasks where context accumulates multiple turns might introduce different dynamics not fully captured here.
- Self-Attribution Bias/Preference Bias/Model Bias, which has been discussed in many previous papers in LLM evaluations, such as ‘https://arxiv.org/abs/2502.06655’ and ‘https://arxiv.org/abs/2410.21819’.  These works are not discussed in thai paper, and the novelty compared with those is unclear.

---

> ### Author Rebuttal · Authors · 2026-03-31
>
> W1:
> To address concerns on realistic agentic tasks, we run this on long-horizon agent-swe-bench, see the figure https://figshare.com/s/9283585a0a1bb9ef32dc.  We also see self-attribution bias in this agentic SWE-bench setting, which is arguably more alarming because that is much closer to a real deployed use case. In addition, we have a breakdown of absolute correctness ratings (0–10) for the original, single step swe-bench: https://figshare.com/s/9ab702ceacb108145635
>
> W2:
> Wataoka et al. (2410.21819) is cited and discussed in our Related Work (Section 2, "LLM Self-Preference in Evaluation").This paper studies only static off-policy settings. The self preference bias paper of two alternative outputs a model chooses when given a comparison task, which the authors hypothesize may be due to models preferring their own style (which can’t be the main cause of our self-attribution bias since we don’t see such an off-policy / on-policy gap using the baseline prompt format).
>
> Chen et al. (2502.06655) discusses a similar phenomenon as Wataoka et al. (which is distinct from what we study), with a focus on potential benchmark contamination. We will add Chen et al. to our related work section. In particular, Chen et al studies a variation of Wataoka et al. in which they use paraphrasing to isolate stylistic preferences (when using baseline prompt formats), which is not the phenomenon we study.
>
> Our paper studies a different phenomenon, how the rating of fixed actions change solely due to implicit authorship attribution, independent of content properties. We hold the artifact constant and vary only whether it appears in an assistant turn (self-attributed) versus a user turn (baseline).
>
> We are happy to add a discussion of this paper to the related work  to clarify the distinction. If you think we have adequately addressed your concerns we would deeply appreciate it if you would consider positively revising your score.

---

### Decision · Program_Chairs · 2026-04-30

**Decision:**

Reject

**Comment:**

The authors study self-attribution bias in LLM agents, showing that models tend to judge their own outputs as more correct and less risky than identical outputs presented without self-attribution. The sentiment mostly shared across reviewers was that this is an important problem, especially given the growing use of LLMs for self-monitoring in agentic systems. The experimental set up is clean and well-controlled, and the results are consistent across domains.

The reviews reflect a mixed assessment. Some reviewers found the empirical findings compelling and emphasized the practical relevance of the identified failure mode. Others raised concerns about novelty relative to prior work on evaluation bias and self-preference, the limited evaluation in more realistic or long-horizon settings, and the lack of deeper mechanistic understanding. Notably, these concerns were not fully resolved during the rebuttal, and the more critical reviewers did not update their scores.

From my reading, the core empirical observation, the gap between on-policy and off-policy evaluation, is interesting and appears to hold in the settings studied in the paper. However, the contribution remains somewhat incremental with respect to existing work on evaluation bias, and the authors do not yet clearly establish a sufficiently distinct conceptual or methodological advance. In addition, while the authors extend the evaluation beyond simple setups, the experimental scope remains limited in terms of capturing more realistic deployment settings, and the lack of deeper analysis of the underlying mechanisms makes it harder to assess the generality and of the phenomenon. The rebuttal provided useful clarifications and additional experiments, which strengthen confidence in the empirical findings. However, key limitations remain, particularly regarding novelty, evaluation breadth, and understanding of the effect. As a result, the paper’s overall contribution relative to the current literature remains somewhat unclear.

I find this paper to be borderline but below the acceptance bar. While the problem is important and the empirical results are suggestive, the current version does not provide a sufficiently strong or well-differentiated contribution to justify acceptance. I recommend rejection.